# 🧙 OF WIKIPEDIA:
# KNOWLEDGE-POWERED CONVERSATIONAL AGENTS

**Emily Dinan**\*, **Stephen Roller**\*, **Kurt Shuster**\*, **Angela Fan, Michael Auli, Jason Weston**
Facebook AI Research
{edinan,roller,kshuster,angelafan,michaelauli,jase}@fb.com

## ABSTRACT

In open-domain dialogue intelligent agents should exhibit the use of knowledge, however there are few convincing demonstrations of this to date. The most popular sequence to sequence models typically "generate and hope" generic utterances that can be memorized in the weights of the model when mapping from input utterance(s) to output, rather than employing recalled knowledge as context. Use of knowledge has so far proved difficult, in part because of the lack of a supervised learning benchmark task which exhibits knowledgeable open dialogue with clear grounding. To that end we collect and release a large dataset with conversations directly grounded with knowledge retrieved from Wikipedia. We then design architectures capable of retrieving knowledge, reading and conditioning on it, and finally generating natural responses. Our best performing dialogue models are able to conduct knowledgeable discussions on open-domain topics as evaluated by automatic metrics and human evaluations, while our new benchmark allows for measuring further improvements in this important research direction.

## 1 INTRODUCTION

Arguably, one of the key goals of AI, and the ultimate the goal of natural language research, is for humans to be able to talk to machines. In order to get close to this goal, machines must master a number of skills: to be able to comprehend language, employ memory to retain and recall knowledge, to reason about these concepts together, and finally output a response that both fulfills functional goals in the conversation while simultaneously being captivating to their human speaking partner. The current state-of-the-art approaches, sequence to sequence models of various kinds (Sutskever et al., 2014; Vinyals & Le, 2015; Serban et al., 2016; Vaswani et al., 2017) attempt to address some of these skills, but generally suffer from an inability to bring memory and knowledge to bear; as indicated by their name, they involve encoding an input sequence, providing limited reasoning by transforming their hidden state given the input, and then decoding to an output. To converse intelligently on a given topic, a speaker clearly needs knowledge of that subject, and it is our contention here that more direct knowledge memory mechanisms need to be employed. In this work we consider setups where this can be naturally measured and built.

We consider the task of open-domain dialogue, where two speakers conduct open-ended chit-chat given an initial starting topic, and during the course of the conversation the topic can broaden or focus on related themes. During such conversations, an interlocutor can glean new information and personal points of view from their speaking partner, while providing similarly themselves. This is a challenging task as it requires several components not found in many standard models. We design a set of architectures specifically for this goal that combine elements of Memory Network architectures (Sukhbaatar et al., 2015) to retrieve knowledge and read and condition on it, and Transformer architectures (Vaswani et al., 2017) to provide state-of-the-art text representations and sequence models for generating outputs, which we term Transformer Memory Networks.

As, to our knowledge, no public domain dataset of requisite scale exists, we build a supervised dataset of human-human conversations using crowd-sourced workers, first crowd-sourcing 1365 diverse discussion topics and then conversations involving $201,999$ utterances about them. Each

---

\*Joint first authors.

topic is connected to Wikipedia, and one of the humans (the *wizard*) is asked to link the knowledge they use to sentences from existing articles. In this way, we have both a natural way to train a knowledgeable conversation agent, by employing a memory component that can recall and ground on this existing text, and a natural way to evaluate models that we build, by assessing their ability at locating and using such knowledge.

Our Transformer Memory Network architectures, both in retrieval and generative versions, are tested in this setup using both automatic metrics and human evaluations. We show their ability to execute engaging knowledgeable conversations with humans, compared to a number of baselines such as standard Memory Networks or Transformers. Our new benchmark, publicly in ParlAI (`http://parl.ai/projects/wizard_of_wikipedia/`), aims to encourage and measure further improvements in this important research direction.

## 2 RELATED WORK

Many existing dialogue tasks do not study the use of knowledge explicitly. For example, popular chit-chat datasets such as Open-Subtitles (Vinyals & Le, 2015), Persona-Chat (Zhang et al., 2018) and Twitter (Sordoni et al., 2015) have tested the ability of sequence-to-sequence models that attend over the recent dialogue history, but do not attempt to recall long-term knowledge beyond encoding it directly into the weights of the feed-forward network.

In the area of goal-directed dialogue, separate from open domain chit-chat, such as airline (El Asri et al., 2017) or restaurant booking (Henderson et al., 2014; Wen et al., 2016; Bordes et al., 2017), knowledge conditioning is typically employed by allowing access to a database through API calls or otherwise. In contrast, our work investigates unstructured knowledge across a large, diverse set of topics potentially spanning all of Wikipedia.

In question answering one does not produce a dialogue response based on a conversation history, but a factual answer based on a question. In that case, it is clear that retrieving and conditioning knowledge is vital. For example, in SQuAD neural models have been developed that attend to a given paragraph from Wikipedia to answer questions (Rajpurkar et al., 2016), or Open-SQuAD which extends this to answering without being given the paragraph, instead performing retrieval over the entirety of Wikipedia (Chen et al., 2017). Recently, the QuAC dataset investigates similar themes, but as a sequence of questions and answers in dialogue form instead (Choi et al., 2018). In this work we do not address question answering, but focus on natural human dialogues which contain a diverse set of utterances, not just questions and answers.

The closest work to ours lies in the area of non-goal directed dialogue incorporating knowledge. The work of Dodge et al. (2016) employed Memory Networks to perform dialogue discussing movies in terms of recommendation and open-ended discussion from Reddit, conditioning on a structured knowledge base. Zhou et al. (2018) also links Reddit to structured knowledge. Both Parthasarathi & Pineau (2018) and Ghazvininejad et al. (2018) use unstructured text instead, as we do: the former to discuss news articles using Wikipedia summaries as knowledge, and the latter to discuss local businesses in two-turn dialogues using Foursquare tips as knowledge. Ghazvininejad et al. (2018) uses an extended Encoder-Decoder where the decoder is provided with an encoding of the context along with the external knowledge encoding. Neither involves dialogue authored with the given knowledge, so it is unclear when knowledge is useful or not. In contrast, in our task, we know the Wikipedia articles and sentences that ground crowdworkers dialogues. Model-wise, Parthasarathi & Pineau (2018) uses a Bag-of-Words Memory Network type fact encoder and an RNN decoder. Our work compares Memory Networks (Sukhbaatar et al., 2015) and Transformers which have been shown to be on-par or superior to RNN encoder-decoders (Vaswani et al., 2017), and develops an architecture that combines these approaches. Concurrently with our work Moghe et al. (2018) proposed a dataset based on the closed domain of movie chats. Our paper shows models working on full multi-turn dialogue in an open-domain setting, which to our knowledge was not shown before.

## 3 WIZARD OF WIKIPEDIA

We consider the following general open-domain dialogue setting: two participants engage in chit-chat, with one of the participants selecting a beginning topic, and during the conversation the topic

is allowed to naturally change. The two participants, however, are not quite symmetric: one will play the role of a knowledgeable expert (which we refer to as the *wizard*) while the other is a curious learner (the *apprentice*).

**Apprentice**    At each stage of the conversation the apprentice talks to the wizard freely, playing the role of a curious learner, eager to chat. Their goal is to go into depth about a chosen topic that interests themselves or their partner, while keeping the conversation engaging and fun. Note that the instruction to delve deeply into a topic makes this different to more "shallow" chit-chat tasks. In this task the use of knowledge is emphasized more.

**Wizard**    The wizard is given the following instructions: *"You have just met the other person, who seems quite curious, and you are eager to discuss a topic with them!"* Their goal is to inform their conversation partner about a topic that one of them will choose. Crucially, the wizard has access to an information retrieval system that shows them paragraphs from Wikipedia possibly relevant to the conversation, which are unobserved by the apprentice. Before each conversation turn the wizard can read these paragraphs and then potentially base their next reply on that observed knowledge. Note, the wizard is particularly instructed not to simply parrot this knowledge, but to use it to craft a relevant reply, and to present any relevant knowledge in a fun and engaging way, if possible.

**Conversation Flow**    The flow of the conversation thus takes place as follows.

1. Either the wizard or apprentice is picked to choose the topic and speak first. The other player receives the topic information, and the conversation begins.

2. When the apprentice sends the wizard a message, the wizard is shown relevant knowledge (described below), and chooses a relevant sentence in order to construct a response, or else chooses the *no sentence used* option.

3. The Wizard responds to the apprentice basing their response on their chosen sentence.

4. The conversation repeats until one of the conversation partners ends the chat (after a minimum of 4 or 5 turns each, randomly chosen beforehand).

After collecting data of such wizard-apprentice conversations between humans, the goal is to then replace the human wizard with a learned agent that will speak to a human apprentice instead, similar to the procedure in Wizard of Oz experiments (Bernsen et al., 2012).

**Topics**    We crowd-sourced a set of 1365 natural, open-domain dialogue topics, each linked to a Wikipedia article. These include diverse topics such as commuting, Gouda cheese, music festivals, podcasts, bowling, and Arnold Schwarzenegger.

**Knowledge Retrieval**    At each step of the dialogue the wizard has access to a set of passages of knowledge which may be relevant to the given dialogue context. While this is a potentially learnable part of the model, we required for this to be fixed so that we could present the results to the annotator when collecting the dataset. We thus used exactly the same retriever that is commonly used for the Open-SQuAD dataset in Chen et al. (2017). It uses a simple inverted index lookup followed by term vector model scoring. Articles and queries are compared as TF-IDF weighted bag-of-word and $n$-gram vectors, using the hashing trick. We retrieve the top 7 articles (first paragraph only) for the last two turns of dialogue (by wizard and apprentice) and the article (first 10 sentences only) for the original topic, and present these articles to the wizard as knowledge context, along with their titles. Note that while this system is used to build the dataset, a superior method can in principle be learned and used by a model at test time.

**Knowledge Selection and Response Generation**    During data collection, the wizard can click on any of the retrieved article titles in the dialogue UI to expand that article, at which point they can click on a sentence that is most relevant to the response they want to make (only one article, and one sentence can be selected on any turn, for simplicity). If they see no relevant article or sentence they can choose *no sentence used* instead. The wizard then enters their response to the apprentice. An image of the Wizard's UI is shown in Appendix A.1.

Table 1: Dataset statistics of the Wizard of Wikipedia task.

| Wizard of Wikipedia Task | Train | Valid | Test Seen | Test Unseen |
|---|---|---|---|---|
| Number of Utterances | 166,787 | 17,715 | 8,715 | 8,782 |
| Number of Dialogues | 18,430 | 1,948 | 965 | 968 |
| Number of Topics | 1,247 | 599 | 533 | 58 |
| Average Turns per Dialogue | 9.0 | 9.1 | 9.0 | 9.1 |
| Knowledge Database | 5.4M articles | | 93M sentences | |

**Final Dialogue Dataset**   The final dialogue dataset we collect consists of 22,311 dialogues with 201,999 turns, which we divide into 166,787 for train, 17,715 for validation, and 17,497 for test. The test set is split into two subsets, Test Seen and Test Unseen. Test Seen contains 533 overlapping topics with the training set with new dialogues about those topics. Test Unseen consists of 58 topics never seen before in train or validation. Overall data statistics can be found in Table 1, and further statistics and examples of collected conversations in Appendix A.2. We observe wizards and apprentices both asking and answering questions, and providing each other with a mixture of facts and personal feelings during their general discussion.

## 4 MODELS

In this work we consider learning dialogue models to replace the *wizard* in our learning tasks, i.e. the knowledgeable speaker. The dialogue model thus can have access to a knowledge source, in this case Wikipedia, to ground the conversation with. We thus develop extensions of the Memory Network (Sukhbaatar et al., 2015) and Transformer (Vaswani et al., 2017) models that can (i) retrieve from a large memory relevant information conditioned on the dialogue history, (ii) carefully read and attend over the retrieved set of knowledge, and then (iii) generate the next dialogue utterance. This model is then used consecutively on each turn to form an entire dialogue with a user.

We develop two classes of models capable of leveraging knowledge: (i) retrieval models that produce an output among a set of candidate responses (the set of utterances from the training set); and (ii) generative models that generate word-by-word (using a beam).

The input to either model is the same: at each dialogue turn where the model is intended to make a response, it is given the current dialogue context $x_1, \ldots, x_t$ of $t$ dialogue turns, where $x_1$ is always the initial starting topic (e.g. "Kurt Cobain"), and the remaining turns swap between the two speakers. The goal at each stage is to output the next utterance $x_{t+1}$.

**Knowledge Retrieval**   We assume a large knowledge base (memory) $m_1, \ldots, m_N$ which is hierarchically organized into documents consisting of paragraphs and sentences. As it is infeasible for current neural attention techniques to operate on this scale, we use standard information retrieval (IR) techniques ($c = \text{IR}(x, m)$) as a first step to return a smaller set of candidates $m_{c_1}, \ldots, m_{c_K}$ for fine-grained selection.

In our experiments, we use the IR system provided to the human annotators during dataset creation, detailed in Section 3. The retriever operates on the topic ($x_1$) and the last two turns ($x_t$ and $x_{t-1}$) if they exist, effectively calling the IR system three times with three different queries. Empirically, this provided better performance compared to merging into one query, likely because it can address quite different topics. We retrieve the top 7 articles (first paragraph only) for each lookup and then flatten all the results into separate sentences (i.e. remove the organization of sentences belonging to articles), but prepend every sentence with its article title. In this way the candidates $m_{c_1}, \ldots, m_{c_K}$ given to the neural model in the next stage can be attended to independently without having to deal with hierarchical issues.

**Knowledge Attention**   We use an attention mechanism to perform fine-grained selection of which knowledge sentences will be used to produce the next turn of dialogue. Each sentence in the memory is independently encoded with a Transformer encoder (Vaswani et al., 2017), and the same Trans-

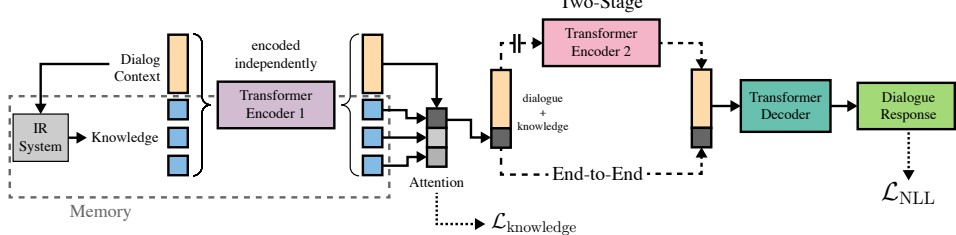

Figure 1: **Generative Transformer Memory Network.** An IR system provides knowledge candidates from Wikipedia. Dialogue Context and Knowledge are encoded using a shared encoder. In the Two-stage model, the dialogue and knowledge are re-encoded after knowledge selection.

former is used to encode the dialogue context $x$. We then perform standard dot-product attention between the memory candidates and the dialogue context.

**Utterance Prediction**    Given the hidden state derived from the memory attention process described above, the final stage is to predict the output utterance that will form the next dialogue turn.

We consider different variants of the two stages above, knowledge attention and utterance prediction, when considering retrieval and generative variants of our models. We will now detail these in turn.

## 4.1   RETRIEVAL TRANSFORMER MEMORY NETWORK

This model encodes each knowledge sentence $m_{c_1}, \ldots, m_{c_K}$ and the dialogue context $x$ with a Transformer, as described above. The final input encoding is calculated by performing dot-product attention over $\mathrm{enc}(m_{c_1}), \ldots, \mathrm{enc}(m_{c_K})$ and adding the resulting weighted sum of these vectors to $\mathrm{enc}(x)$ to get the representation $\mathrm{rep}_{\mathrm{LHS}}(m_{c_1}, \ldots, m_{c_K}, x)$. The candidate responses $r_1, \ldots, r_L$ are encoded with a separate Transformer to get $\mathrm{rep}_{\mathrm{RHS}}(r_i)$ for each $i$. We choose as a response $r_\ell$ where

$$\ell = \underset{i \in \{1, \ldots, L\}}{\arg\max} \frac{\mathrm{rep}_{\mathrm{LHS}}(m_{c_1}, \cdots, m_{c_K}, x)}{\|\mathrm{rep}_{\mathrm{LHS}}(m_{c_1}, \ldots, m_{c_K}, x)\|_2} \bullet \frac{\mathrm{rep}_{\mathrm{RHS}}(r_i)}{\|\mathrm{rep}_{\mathrm{RHS}}(r_i)\|_2}.$$

The model is trained to minimize the cross-entropy loss, where the negative candidates for each example are the responses to the other examples in the batch (Henderson et al., 2017).

## 4.2   GENERATIVE TRANSFORMER MEMORY NETWORK

We consider two versions: a Two-stage and an End-to-end version. Both models find the most relevant piece of knowledge $m_{\mathrm{best}}$, and then perform an encoding step by concatenating it with the dialogue context, allowing the decoder to attend over both the knowledge and dialogue when formulating its response. We employ a beam search of 5 to select our best response. All generative models employ BPE encoding (Sennrich et al., 2016), which we found effective at enabling generators to copy rare words from Wikipedia sentences (Fan et al., 2018).

In the **End-to-end** version, a shared Transformer encoder is used to encode all candidates $m_{c_i}$ and the dialogue history. The encoded candidates are flattened into vectors using the normalization from Cer et al. (2018) (summing, and normalizing by the square root of the sentence length in order to balance short and long sentences) to produce an attention prediction over the memory. The full sequence encoding of the single highest selected knowledge $m_{\mathrm{best}}$ is concatenated with the encoding of the dialogue, and passed into a Transformer decoder. An illustration of our End-to-end model is shown in Figure 1. We train the model to minimize the negative log-likelihood of the response utterance. We can add additional supervision by forcing the knowledge selection to correctly choose the same knowledge candidate as the human wizard in the training set by adding an additional cross-entropy loss over the knowledge attention, modulated by a weight $\lambda$:

$$\mathcal{L} = (1 - \lambda)\mathcal{L}_{\mathrm{NLL}} + \lambda\mathcal{L}_{\mathrm{knowledge}}.$$

In the **Two-stage** version, we employ two separately trained models for each of these two tasks, knowledge selection and utterance prediction. As the knowledge selection step creates a hard deci-

Table 2: **Test performance of various methods on the Knowledge Selection Task.** The models must select the gold knowledge sentences chosen by humans given the dialogue context.

| | Seen Test | | Unseen Test | |
|---|---|---|---|---|
| Method | R@1 | F1 | R@1 | F1 |
| Random | 2.7 | 13.5 | 2.3 | 13.1 |
| IR baseline | 5.8 | 21.8 | 7.6 | 23.5 |
| BoW MemNet | 23.0 | 36.3 | 8.9 | 22.9 |
| Transformer | 22.5 | 33.2 | 12.2 | 19.8 |
| Transformer (+Reddit pretraining) | 24.5 | **36.4** | **23.7** | **35.8** |
| Transformer (+Reddit pretraining, +SQuAD training) | **25.5** | 36.2 | 22.9 | 34.2 |

sion influencing the output of the generator, we find maximizing the performance of this component to be vital. We can also improve performance of the decoder by employing *knowledge dropout* (K.D.), wherein we artificially prevent the model from attending to knowledge a fraction of the time during training. We find this helps the generator be more resilient to errors at the knowledge selection stage, and makes training faster. K. D. is a novel technique we propose here, however it is similar to many other dropout techniques, e.g. feature dropout used in Wu et al. (2017).

## 5 EXPERIMENTS

We describe each of our experimental setups and results. We first investigate the ability of our models to select knowledge appropriately, and then consider the full task of dialogue with knowledge.

### 5.1 KNOWLEDGE SELECTION TASK

Before looking at the full Wizard dialogue task, we assess the ability of models to predict the knowledge selected by human wizards in the dataset given the dialogue history. This will inform us of the feasibility of this task and the best models to use in a two-stage architecture. We compare Transformers against various baselines including a random baseline; an Information Retrieval (IR) baseline, which uses simple word overlap; and a Bag-of-Words Memory Network (Sukhbaatar et al., 2015). Where noted, the Transformer is pretrained on Reddit data (Mazaré et al., 2018), and fine-tuned for our task. The results are shown in Table 2. Transformers work best, as long as they are pretrained on a large dataset (Reddit), while multi-tasking on SQuAD provides marginal impact. Further analysis of this task using other models is provided in Appendix B.1. We use the best performing Transformer model reported here for our two-stage generative Memory Network in the full dialogue task.

### 5.2 FULL TASK: DIALOGUE WITH KNOWLEDGE

We evaluate our models on the full task of dialogue generation given knowledge in two settings: given the gold knowledge sentence chosen by a human, or where the model needs to predict which knowledge to use. We separately describe experiments for retrieval and generative models.

**Retrieval Experiments**  We use similar baselines as in the knowledge selection experiments, but now also apply Transformer Memory Networks, which attend over knowledge. Models are evaluated measuring Recall@1 when ranking the gold response among 99 randomly chosen candidates, and unigram F1 of the model's prediction with the gold response. The results are shown in Table 3. We find that the addition of knowledge improves all models (improving Bow MemNet from 56 to 71 R@1 and the Transformer MemNet from 79 to 87 R@1) for predicted knowledge. Performance improves dramatically when models are provided gold knowledge, but otherwise retain similar trends.

**Generative Experiments**  We compare our generative End-to-end and Two-stage Transformer Memory Network models to two more baselines: repeating the last utterance, and a generative Transformer model trained to respond to dialogue but without access to knowledge. Models are evaluated using perplexity (PPL) of the gold response and unigram F1.

Table 3: **Retrieval methods on the full Wizard task.** Models must select relevant knowledge and retrieve a response from the training set as a dialogue response. Using knowledge always helps, and the Transformer Memory Network with pretraining performs best.

| | Predicted Knowledge | | | | Gold Knowledge | |
| | Test Seen | | Test Unseen | | Seen | Unseen |
| Method | R@1 | F1 | R@1 | F1 | R@1 | R@1 |
|---|---|---|---|---|---|---|
| Random | 1.0 | 7.4 | 1.0 | 7.3 | 1.0 | 1.0 |
| IR baseline | 17.8 | 12.7 | 14.2 | 11.6 | 73.5 | 67.5 |
| BoW MemNet (no knowledge) | 56.1 | 14.2 | 28.8 | 11.6 | 56.1 | 28.8 |
| BoW MemNet | 71.3 | **15.6** | 33.1 | 12.3 | 84.5 | 66.7 |
| Transformer (no knowledge, w/o Reddit) | 60.8 | 13.3 | 25.5 | 9.7 | 60.8 | 25.5 |
| Transformer (no knowledge, w/ Reddit) | 79.0 | 15.0 | 54.0 | 11.6 | 79.0 | 54.0 |
| Transformer MemNet (w/ Reddit) | 86.8 | 15.4 | **69.8** | **12.4** | 91.6 | 82.3 |
| Transformer MemNet (w/ Reddit+SQuAD) | **87.4** | 15.4 | **69.8** | **12.4** | **92.3** | **83.1** |

Table 4: **Generative models on the full Wizard Task.** The Two-stage model performs best using predicted knowledge, while the End-to-end (E2E) model performs best with gold knowledge.

| | Predicted Knowledge | | | | Gold Knowledge | | | |
| | Test Seen | | Test Unseen | | Test Seen | | Test Unseen | |
| Method | PPL | F1 | PPL | F1 | PPL | F1 | PPL | F1 |
|---|---|---|---|---|---|---|---|---|
| Repeat last utterance | - | 13.8 | - | 13.7 | - | 13.8 | - | 13.7 |
| Transformer (no knowledge) | - | - | - | - | 41.8 | 17.8 | 87.0 | 14.0 |
| E2E Transformer MemNet (no auxiliary loss) | 66.5 | 15.9 | 103.6 | 14.3 | 24.2 | 33.6 | 35.5 | 29.5 |
| E2E Transformer MemNet (w/ auxiliary loss) | 63.5 | 16.9 | 97.3 | 14.4 | **23.1** | **35.5** | **32.8** | **32.2** |
| Two-Stage Transformer MemNet | 54.8 | 18.6 | 88.5 | **17.4** | 30.0 | 30.7 | 42.7 | 28.6 |
| Two-Stage Transformer MemNet (w/ K.D.) | **46.5** | **18.9** | **84.8** | 17.3 | 28.6 | 30.6 | 43.7 | 28.0 |

Table 5: **Human Experiments.** Evaluations of the best generative and retrieval models on full dialogues with humans. Human ratings are reported as mean (stddev). Wiki F1 measures unigram overlap with the Wikipedia entry for the chosen topic, a measure of knowledge used in conversations.

| | Seen | | Unseen | |
| Method | Rating | Wiki F1 | Rating | Wiki F1 |
|---|---|---|---|---|
| Human Performance | 4.13 (1.08) | 11.1 | 4.34 (0.98) | 10.6 |
| Retrieval Transformer (no knowledge) | 3.33 (1.30) | 19.8 | 3.12 (1.34) | 13.7 |
| Generative Transformer (no knowledge) | 2.11 (1.11) | 15.3 | 2.54 (1.38) | 10.1 |
| Retrieval Transformer MemNet | 3.43 (1.10) | 23.4 | 3.14 (1.31) | 16.3 |
| Two-Stage Generative Transformer MemNet | 2.92 (1.33) | 30.0 | 2.93 (1.30) | 26.2 |

The results are given in Table 4. Our experiments show that both the End-to-end and Two-stage models employ the knowledge in their response predictions, as they outperform their counterpart Transformer without knowledge, and demonstrate substantial improvements when provided the gold knowledge. While the Two-stage model produces significantly stronger perplexity and F1 scores using the predicted knowledge, the End-to-end model outperforms the Two-stage model in the gold knowledge experiments. This suggests that the Two-stage model benefits from the strong knowledge selection module (Section 5.1), but that the End-to-end model is better at employing the selected knowledge. Furthermore, we find that the additional knowledge selection supervision (auxiliary loss) in the End-to-end model improves it on every metric, suggesting that tightly integrating these tasks is beneficial. Knowledge dropout (K. D.) also helps (compare last two rows). More evidence

| **Model:** Retrieval Transformer MemNet |
| --- |

**Topic:** E-book (*seen*)
Human: i do not know why, but I have never gotten into E-books
Model: Same here. I have a kindle, but I don't use it for books. It's nicer to hold a real book and turn pages.
Human: yes, i like the physical feel and smell of a real book
Model: I actually had a Nook e-book where you had to download the e-book after selecting it online. I grew out of it, I prefer paper books also.
Human: i tried e-books but I do not like reading on a screen, its disorientated for me
Model: Yes, reading a book on a tablet just isn't the same!
Human: plus I like owning the physical copy of a real book, I feel like I own it
Model: I prefer print and newsreels.

**Topic:** Toga party (*seen*)
Model: I love toga parties! I usually make my toga out of a bed sheet and wear my sandals.
Human: Me too!!! I usually use some flip flops and my blanket!
Model: Toga parties typically have games involving kegs and excessive drinking!
Human: Yeah! And getting to wear funny outfits and a crown of leaves!
Model: I love me some Toga Parties!
Human: I love me some too!
Model: Toga parties are so much fun with the Roman or Greek theme.
Human: Yeah! I especially like Greek themes

| **Model:** Two-Stage Generative Transformer MemNet |
| --- |

**Topic:** Shower (*unseen*)
Human: So I've been looking for a better showerhead with more water pressure for my shower.
Model: I love pressure-versed water at nearly constant temperature to a shower
Human: That sounds pretty great! Love those rainshower showers!
Model: the ancient greeks were the first people to have showers.
Human: Well I knew they were good for something.
Model: the original showers were neither indoor structures nor man made but were common.
Human: Ah that sounds intimidating!
Model: it is required manual transport of both fresh and waste water.

Figure 2: Selected conversations between humans and models.

for this is shown in Appendix B.1. Lastly, we note that both Two-stage models give higher F1 scores than any of the retrieval models shown in Table 3.

## 5.3 HUMAN EVALUATION

We perform human evaluation of our models using crowd-sourced workers. Humans are paired with our models and asked to chat about a specific topic (given a choice of 2–3 topics) for 3–5 dialogue turns. Following their conversation, the humans are asked to rate their dialogue partner on a scale of 1–5, with the rating indicating how much they "liked" the conversation (5 is best), which we refer to as the engagingness rating. Using the collected conversations, we also calculate a metric we call the Wiki F1 sore: the F1 overlap of the model's utterances with the first 10 sentences of the Wikipedia page for the chosen topic as a proxy for how much knowledge the model exhibits. We seek a model that can be simultaneously engaging and knowledgeable, hence we would like to maximize both these metrics[1]. For comparison, we also collect 100 human-human conversations, with only one human choosing the topic and performing evaluation. In total, we collect a total of 546 conversations with ratings from 464 distinct workers. These results are shown in Table 5.

We find that the retrieval models significantly outperform the generative models on the human engagingness evaluation(Student's t-test, $p < .05$). The human engagingness differences between retriever models with and without knowledge are not significant, but note they both trend toward use of knowledge due to the candidate sentences retrieved, with the knowledgeable version obtaining significantly higher Wiki F1 scores in both seen and unseen test sets.

For the generative models, we find human engagingness ratings are significantly improved by the use of knowledge ($p < .01$). The significantly higher Wiki F1 scores indicate that (i) these models convey more knowledge than their counterparts without knowledge conditioning; and (ii) on both seen and unseen sets they convey more knowledge than the retrieval models. In particular, on unseen

---

[1]For example, a model could display knowledge by copying parts of Wikipedia, but not be engaging at all.

data the gap between retrieval and generative models is larger. This is understandable, as retrieval models are limited to producing a response from the training set where the unseen topic did not appear.

There is still a considerable gap to human ratings of other humans compared to all our models (first row of Table 5). Figure 2 shows example conversations with the retrieval and generative models. Additional analysis and examples can be found in Appendix B.3 and C.

## 6 CONCLUSION

In this work we build dialogue agents which are able to employ large memory systems containing encyclopedic knowledge about the world in order to conduct engaging open-domain conversations. We develop a set of architectures, Transformer Memory Network models, that are capable of retrieving and attending to such knowledge and outputting a response, either in retrieval or generative modes. To train and evaluate such models, we collect the Wizard of Wikipedia dataset, a large collection of open-domain dialogues grounded by Wikipedia knowledge, and demonstrated the effectiveness of our models in automatic and human experiments. Our new publicly available benchmark aims to encourage further model exploration, and we expect such efforts will result in significant advances in this important research direction.

There is much future work to be explored using our task and dataset. Some of these include: (i) bridging the gap between the engagingness of retrieval responses versus the ability of generative models to work on new knowledge and topics, (ii) learning to retrieve and reason simultaneously rather than using a separate IR component; and (iii) investigating the relationship between knowledge-grounded dialogue and existing QA tasks which also employ such IR systems. The aim is for those strands to come together to obtain an engaging and knowledgeable conversational agent.

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

# A    DATASET COLLECTION

## A.1    HUMAN ANNOTATION INTERFACE (FOR WIZARD)

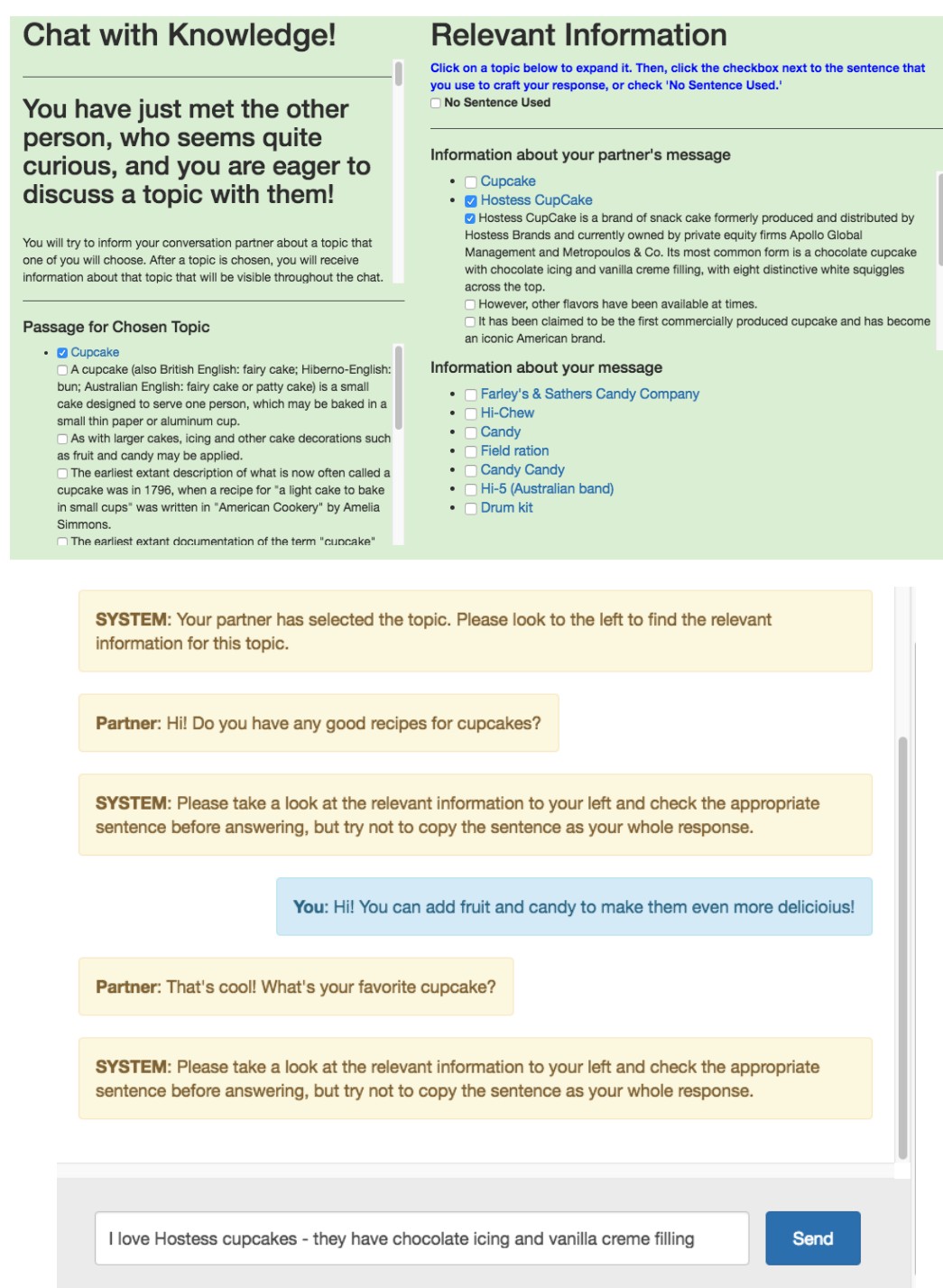

## A.2   WIZARD OF WIKIPEDIA EXAMPLES

| Topic: | Lifeguard |
|---|---|
| Apprentice: | So I am a lifeguard. Know anything about saving lives in water? |
| Wizard: | I'm impressed! It's a big responsibility to supervise other people's safety in the water! Tell me more. |
| Apprentice: | Well, I help make sure people do not drown or get injured while in or near the water! |
| **Knowledge:** | A lifeguard is a rescuer who supervises the safety and rescue of swimmers, surfers, . . . Lifeguards are strong swimmers and trained in CPR/AED first aid, certified in water . . . 
 . . . 
 In some areas, the lifeguard service also carries out mountain rescues, or may function as the primary EMS provider. |
| Wizard: | I've heard that in some places, lifeguards also help with other sorts of emergencies, like mountain rescues! Is that part of your job too? |
| Apprentice: | I have! I feel like you know much about this! What brings you to know so much? |
| Wizard: | Oh, that's about the extent of my knowledge. I've just been around beaches and I've always admired lifeguards. I'm not a super strong swimmer myself. |
| Topic: | Armadillo |
| Wizard: | I love animals and think armadillos are awesome with their leathery shell. |
| Apprentice: | I don't think I've ever seen an armadillo in real life! |
| Wizard: | I've seen them at the zoo. Armadillo means little armored one in Spanish. |
| Apprentice: | Are they native to a Spanish-speaking part of the world? |
| **Knowledge:** | Armadillos are New World placental mammals in the order Cingulata . . . The word "armadillo" means "little armoured one" in Spanish. 
 . . . 
 The nine-banded armadillo ("Dasypus novemcinctus"), or the nine-banded, long-nosed armadillo, is a medium-sized mammal found in North, Central, and South America. |
| Wizard: | Yes, they are most commonly found in North, Central, and South America |
| Topic: | Ice cream |
| Wizard: | I just love ice cream. I love the types with fruits and flavours. Do you like ice cream? |
| Apprentice: | I love Ice cream as much as any one. I especially like Gelato, foreign ice cream! |
| **Knowledge** | Ice cream is a sweetened frozen food typically eaten as a snack or dessert.. . . It is usually made from dairy products, such as milk and cream, and . . . 
 . . . 
 Bacon ice cream (or bacon-and-egg ice cream) is an ice cream generally created by adding bacon to egg custard and freezing the mixture. |
| Wizard: | Me too. There are some strange combinations though, have you heard of bacon ice cream? where they add bacon and even egg custard to the freezing mixture! |
| Apprentice: | Surprisingly bacon ice cream doesn't surprise me. That doesn't sound appealing to me, but perhaps it could be delicious. . . |

Figure 3: **The Wizard of Wikipedia dataset.** Examples of collected conversations from the dataset, where both wizard and apprentice are humans. The wizard has access to an information retrieval system over Wikipedia, so that they can ask and answer questions, and make statements relevant to the discussion. For each utterance, knowledge retrieval is performed based on dialogue history, giving ∼61 knowledge candidates per turn, with wizards clicking *no sentence used* 6.2% of the time. Assuming that a question contains a question mark or begins with 'how', 'why', 'who', 'where', 'what' or 'when', in the dataset Apprentices ask questions in 13.9% of training set utterances, and answer questions (i.e., the Wizard has asked a question) 39.5% of the time, while saying new or follow-on statements (neither asking nor answering a question) 49.3% of the time. Hence, the wizard and apprentice conduct conversations with a variety of dialogue acts.

## A.3   TOPIC SELECTION

To choose between topics that are natural we employed the existing Persona-Chat dataset (Zhang et al., 2018) where crowdworkers where asked to create personas of typical speakers. There are

~1000 personas, each of which consists of 4-5 sentences describing that person's interests, e.g. "I love watching Game of Thrones", "I like to eat cheetos" and "I recently got a cat". These can thus naturally be seen as topics of interest, and using another set of annotators we mapped each sentence to 1 or more relevant Wikipedia pages, if possible, e.g. "Ariel is my favorite princess" was labeled with the Wikipedia page for *The Little Mermaid*. As some sentences are harder to connect with Wikipedia, e.g. "I am witty", they are left unlabeled. We thus obtain 1,431 topics in total to use for our task. We retain the persona topic sets and thus present 2-3 related topic choices as conversation starters per dialogue during data collection.

# B  ADDITIONAL EXPERIMENTS

## B.1  KNOWLEDGE SELECTION

Table 6: **Test performance of the Knowledge Selection Tasks.** We also tested the performance of our models trained to do the full dialogue task (see Section 5.2) on the knowledge selection task. For our retrieval system, this refers to the performance of the knowledge attention. The results show that our retrieval system could be improved, and the auxiliary loss clearly helps the generative models.

| | Seen Test | | Unseen Test | |
| --- | --- | --- | --- | --- |
| Method | R@1 | F1 | R@1 | F1 |
| Random | 2.7 | 13.5 | 2.3 | 13.1 |
| IR baseline | 5.8 | 21.8 | 7.6 | 23.5 |
| BoW MemNet | 23.0 | 36.3 | 8.9 | 22.9 |
| Transformer | 22.5 | 33.2 | 12.2 | 19.8 |
| Transformer (+Reddit pretraining) | 24.5 | **36.4** | **23.7** | **35.8** |
| Transformer (+Reddit pretraining, +SQuAD training) | **25.5** | 36.2 | 22.9 | 34.2 |
| Retrieval Transformer MemNet (no auxiliary loss) | 12.9 | 24.6 | 14.6 | 26.3 |
| Generative E2E Transformer MemNet (no auxiliary loss) | 13.4 | 28.3 | 11.8 | 25.9 |
| Generative E2E Transformer MemNet (w/ auxiliary loss) | 21.1 | 32.8 | 14.3 | 22.8 |

## B.2  FULL DIALOGUE WITH RETRIEVAL

## B.3  HUMAN EXPERIMENTS

# C  ERROR ANALYSIS

We perform an analysis of the dialogues produced from the human evaluation experiments detailed in Section 5.3. We sample 20 conversations from each experimental setting, split between *seen* and *unseen*. Conversations are re-tokenized and lowercased to reduce superficial differences between models, and then analyzed in a single-blind setup. We note of common errors and behaviors exhibited in each of the different conversations.

In general, the human-human conversations are starkly different than any of the bot conversations – humans tend to have more small talk, or use the topic of discussion as a mere icebreaker, with neither human behaving as a wizard. This is in contrast to human-human conversations from the Wizard dataset itself, where one human has access to Wikipedia, and the conversation becomes more grounded in factual sentences. Similarly, all models attempt to play the role of wizard and produce more factual sentences too. In some rounds, humans treat the bot as a sort of question-answer machine, suggesting that the models could be improved by additionally employing SQuAD-like training data.

The retriever *without* knowledge is particularly prone to non sequiturs, or rapidly changing the subject. During unseen conversations, it is especially likely to discuss something other than the chosen topic. In contrast, the retriever *with* knowledge tends to stick to the chosen topic strongly, but has difficulty if the human changes the subject. Frequently in unseen topics, the retriever with

Table 7: **Retrieval methods on the full Wizard task.** In addition to the models we tested in the paper, we also tested a two-stage retrieval system in which we used our best-performing model on the knowledge selection task to choose a single knowledge sentence to condition on for the dialogue retrieval task. This outperformed our best retrieval method in terms of F1 but not not in terms of Recall@1. Furthermore, we compared these results to a two-stage retrieval system in which the dialogue retrieval module is optimized for seeing the gold chosen knowledge sentence. The performance of this system on the gold knowledge task suggests that the retrieval system could be improved by increasing performance on the knowledge selection subtask.

| | Predicted Knowledge | | | | Gold Knowledge | |
| | Test Seen | | Test Unseen | | Seen | Unseen |
| Method | R@1 | F1 | R@1 | F1 | R@1 | R@1 |
|---|---|---|---|---|---|---|
| Random | 1.0 | 7.4 | 1.0 | 7.3 | 1.0 | 1.0 |
| IR baseline | 17.8 | 12.7 | 14.2 | 11.6 | 73.5 | 67.5 |
| BoW MemNet (no knowledge) | 56.1 | 14.2 | 28.8 | 11.6 | 56.1 | 28.8 |
| BoW MemNet | 71.3 | 15.6 | 33.1 | 12.3 | 84.5 | 66.7 |
| Transformer (no knowledge, w/o Reddit) | 60.8 | 13.3 | 25.5 | 9.7 | 60.8 | 25.5 |
| Transformer (no knowledge, w/ Reddit) | 79.0 | 15.0 | 54.0 | 11.6 | 79.0 | 54.0 |
| Transformer MemNet (w/ Reddit) | 86.8 | 15.4 | **69.8** | 12.4 | 91.6 | 82.3 |
| Transformer MemNet (w/ Reddit+SQuAD) | **87.4** | 15.4 | **69.8** | 12.4 | 92.3 | 83.1 |
| Two-stage Transformer (optimized for predicted knowledge) | 84.2 | 16.2 | 63.1 | **13.2** | 92.3 | 83.1 |
| Two-stage Transformer (optimized for gold knowledge) | 79.6 | **16.6** | 60.1 | 13.1 | **96.3** | **88.3** |

Table 8: **Human Experiments.** We calculate the Wiki F1 score for the wizard and apprentice as they appear in the dataset for the sake of comparison to our human evaluations. Note that this differed from the human-human evaluation set-up in the sense that the wizard had direct access to Wikipedia passages in the UI, which explains the higher values of Wiki F1 both for the wizard (who uses Wikipedia) and for the apprentice (who would likely reference that use).

| | Seen | | Unseen | |
| Method | Rating | Wiki F1 | Rating | Wiki F1 |
|---|---|---|---|---|
| Human Performance | 4.13 (1.08) | 11.1 | 4.34 (0.98) | 10.6 |
| Wizard Performance (in dataset) | - | 43.3 | - | 43.1 |
| Apprentice Performance (in dataset) | - | 23.2 | - | 23.7 |
| Retrieval Transformer (no knowledge) | 3.33 (1.30) | 19.8 | 3.12 (1.34) | 13.7 |
| Generative Transformer (no knowledge) | 2.11 (1.11) | 15.3 | 2.54 (1.38) | 10.1 |
| Retrieval Transformer MemNet | 3.43 (1.10) | 23.4 | 3.14 (1.31) | 16.3 |
| Two-Stage Generative Transformer MemNet | 2.92 (1.33) | 30.0 | 2.93 (1.30) | 26.2 |

knowledge produces similar, but factually inaccurate answers to user queries. For example, when one user asks about parts of Ireland to visit, the model enumerates a list of locations in Greece. Nonetheless, its repertoire of available responses often include inviting responses, allowing the bot to have a more natural conversational flow. Selected conversations with the retriever with knowledge may be found in Figure 4, for both seen and unseen topics.

The generator *without* knowledge is particularly prone to many of the typical behaviors of seq2seq systems (Li et al., 2016; Vijayakumar et al., 2016), including local repetition ("cookies are made of flour, oil, oil, and oil"), global repetition (producing the near same utterance for multiple turns), or inconsistencies in its personality (saying it both likes and dislikes movies). The generator with knowledge has significantly fewer issues with repetition, as it errs on the side of copying large fragments from the Wikipedia knowledge. The generator *with* knowledge can also act as a selfish conversationalist, choosing to respond or detail information without inviting a response. Although it generally produces accurate statements, it sometimes produces statements using an incorrect date,

name or word. It also frequently produces formulaic responses, like "I don't know, but I do know that [Wikipedia excerpt]". Nonetheless, we find the generator with knowledge is able to successfully generalize to unseen topics using the knowledge from Wikipedia. Selected conversations with the generator with knowledge may be found in Figure 5.

---

**Model:** Retrieval Transformer MemNet

---

**Topic:** Arnold Schwarzenegger (*seen*)
Human: Hey are you into the movie The Running Man
Model: Yeah! That's one of my favorite movies! I can't believe that Team America: World Police was made in 2004 - 14 years ago!
Human: Another one of Arnies movies I like is Terminator 2 JUDGMENT DAY
Model: The Terminator is probably his most iconic role, but he got his start from the movie "Conan the Barbarian", which came out a couple years earlier. I actually haven't seen that one.
Human: Which one? Conan or "Da Terminata"
Model: I like Arnold Schwarzeneggar as well! I thought he was a fantastic governator, serving from 2003-2011.
Human: HAha Governator! Yeah, it's pretty crazy they he held higher office.
Model: Yes, he won the LIFETIME ACHIEVEMENT AWARD! Interestingly, he joined the U.S. Army in 1961 and was a paratrooper! *(*)*

---

**Topic:** Steak (*seen*)
Human: Hello, it's lunch time here, and I'm in the mood for a great steak
Model: Wow that sounds delicious I wish I could say the same hahah, When the word "steak" is used without qualification, it normally refers to a Beefsteak
Human: Interesting, I don't really have a favorite steak, how about you?
Model: I like rib eye it is really good specially medium rare, When the word "steak" is used without qualification, it generally refers to a beefsteak
Human: I like rib eye too! What is it about a rib eye that you enjoy?
Model: Most steak cuts are very tender meat but I think rib eye can be quite tender
Human: They really are. How do you choose a good steak?
Model: When it comes to steak and other cuts of meat you have rare, medium rare, medium, well done and so on
Human: How do you prefer your steak prepared?
Model: I like that you can prepare it many different ways. You can fried it, grill it, or even broil the steak

---

**Topic:** Ireland (*unseen*)
Model: Hi, have you ever been to Dublin? It is the capital and largest city in Ireland.
Human: No, I haven't but would love to
Model: It is the largest city in Northern Ireland. *(*)*
Human: I've always found Ireland to be fascinating and would love to visit sometime
Model: Same here! It is the capital of Ireland too!
Human: What is the most popular place to visit in Ireland?
Model: I would recommend the Thrace, Crete, and the Ionian Islands as must see places. *(*)*
Human: I don't know anything about the Ionian Islands
Model: It was once referred to as the Hellenic Republic! *(*)*

---

**Topic:** American football (*unseen*)
Model: American football can be traced back to Britain in the mid-19th century. The rules were very similar!
Human: I did not know that. Did they wear equipment back then?
Model: Am not really certain about it but American football resulted from several major divergences from association football and rugby football
Human: The Philadelphia eagles actually have a rugby player on their team this year. he never played football until this year.
Model: Philadelphia is one of a few cities to have all four teams, but they love the Eagles more than anything else by a long shot!
Human: As a life long Philadelphia sports fan I have to agree with you.
Model: I've always been more of a fan of the American Football team from Pittsburgh, The Steelers!
Human: Being from Philadelphia I am not much of a pittsburgh fan.
Model: Are the Pittsburgh Steelers in the AFC?
Human: Yes they are. the AFC north to be exact.

---

**Topic:** List of national parks of the United States (*unseen*)
Human: I have never visited a national park in the United States, but I would really like to!
Model: Me too. I'd love to live in a national park
Human: Which one do you think is the best to visit first?
Model: The United Nations is an important place to visit, the Statue of Liberty, Central park is great also
Human: Is the Statue of Liberty considered a National Park?
Model: Yes, the statue of Liberty is there on an island - The Liberty Island.
Human: I had no idea that that was a National Park. I have heard Zion National Park is really nice!
Model: Yes, it links the shenandoah national park and the great smoky mountains together! *(*)*

---

Figure 4: **Retriever with Knowledge conversations**. Selected conversations with a human. *(*)* indicates clear factual mistakes by the model.

---

**Model:** Two-Stage Generative Transformer MemNet

---

**Topic:** Circus (*seen*)

Model: have you ever been to a circus? it is a company of performers who put on diverse entertainment shows.

Human: I went to the circus once when I was a child actually.

Model: that's great! entertainment is a form of activity that holds the attention and interest of an audience.

Human: That's true. Have you ever been to a circus?

Model: I have not, but I know they were built with various types of seating ring, and sometimes a stage.

Human: I went to one held in a stadium, so we had box seats.

Model: that's awesome! i'm sure you'll have to go to the circus, with appropriate music.

Human: Yes, circuses usually have their own kind of music.

Model: circuses can be done by circuses.

---

**Topic:** Granny Smith (*seen*)

Human: I really love Granny Smith apples, they're my favorite type of apple.

Model: I love granny smith apples. they have hard, light green skin and a crisp flesh.

Human: Yes, I really enjoy them. I also like Honeycrisp apples but they're so expensive!

Model: they've been grown for thousands of years in asia and europe, and were brought to north america by european colonists *(*)*

Human: Oh really? They've been around way longer than I thought!

Model: they're also consumed raw, it's one of the most popular cooking apples.

Human: I always eat them raw, I hate cooked apples for some reason, even in pie.

Model: i'm not sure if it's cooked, but I do know that they can be cooked raw.

---

**Topic:** Culture of Chicago (*seen*)

Human: hi do you like chicago

Model: I love chicago. it's the third most populous city in the us.

Human: how many people live there?

Model: about 2. 7 million residents, it is also the most populous city in the state of illinois.

Human: do you like chicago deep dish pizza?

Model: chicago-style pizza may be made in deep dish style and as a stuffed pizza.

Human: what is the name of the baseball team?

Model: dano's pizzeria specializes in chicago style stuffed pizza.

Human: Is Dano's pizzaria open everyday?

Model: yes it is. it is an american style pizza chain headquartered in midtown brook, illinois.

---

**Topic:** Animals in sport (*unseen*)

Model: i love to watch animals. it's the most popular form of animal-related sport.

Human: horse racing is the most awesome animal sport!

Model: horse racing is an equestrian performance sport, typically involving two or more horses ridden by jockeys or driven over a set distance for competition.

Human: yes, the Kentucky Derby and the triple crown is something I look forward to every year!

Model: the horse has evolved over the past 45 to 55 million years from a small multi toed creature

Human: that is very interesting, how did people come to domesticate horses for sport?

Model: i'm not sure but the domestication is believed to have been widespread by 3000 bc

Human: very interesting! I wonder how long people used horses to hunt before deciding to race them.

Model: i'm not sure but I do know that horses have been domesticated for thousands of years.

---

**Topic:** Dairy farming (*unseen*)

Human: hi there i was wondering how you felt about farming

Model: I do not like dairy farming but I do know that in developed countries, dairy farms typically consist of high dairy products.

Human: oh yes they have alot of cows and push out alot of dairy every hour

Model: other species used in commercial dairy farming include goats, sheep, and camels

Human: i did not know you could get milk a camel

Model: camel milk has supported bedouin

Human: oh wow thats very interesting

Model: in italy they are growing in popularity to produce an alternative milk source for humans

Human: i have had goat milk before

Model: the earliest evidence of using domesticated cows for dairy production is the seventh millennium bc-the neolithic era -

Human: thats so neat. i would say cow milk is my favorite

Model: I love cow milk, especially since the iphone is used in the iphone.

Human: iphone and cow milk

Model: the iphone was first released in 2007.

Human: oh ok well thanks for the talk have a great day

Model: no problem! the iphone was first released in 2007.

---

Figure 5: **Two-stage Generator conversations.** Selected conversations with a human. *(*)* indicates clear factual mistakes by the model.

