# OpenReview forum: "Wizard of Wikipedia: Knowledge-Powered Conversational Agents"
_ICLR.cc/2019/Conference_

### Official Review · AnonReviewer1 · 2018-11-01
**Interesting new dataset**

**Rating:** 8
**Confidence:** 4

**Review:**

This paper introduces a new dataset and method for chatbots. In contrast to previous work, this paper specifically probes how well a dialogue system can use external unstructured knowledge.

Quality:
Overall, this is a very high-quality paper. The dataset is developed well, the experimental setup is well thought-through and the authors perform many ablation studies to test different model variants. The main criticism I have would be that the human evaluation is rather simple (rating 1-5), I would have expected more fine-grained categories, especially ones that relate to how much knowledge the system uses (I appreciate the "Wiki F1" metric, but that is an automatic metric). As it is, the human evaluation shows that most of their contributions are not appreciated by human annotators. Further, the paper ends a bit abruptly, I would have expected a more in-depth discussion of next steps.

Clarity:
The description of the work is clear in most places. I particularly like the abstract and introduction, which set up the rest of the paper nicely. In some places, perhaps due to space restrictions, method descriptions are a bit too short.

Originality:
The paper is fairly original, especially the aspect about specifically using external knowledge. The authors could have been more clear on how the work differs from other work on non-goal directed dialogue work though (last paragraph of related work section).

Significance:
The dataset is really well-developed, hence I believe many working in the dialogue systems community will re-use the developed benchmark and build on this paper.

More detailed comments:
- Missing reference for goal-oriented dialogue datasets: Wen et al. 2017, A Network-based End-to-End Trainable Task-oriented Dialogue System, https://arxiv.org/abs/1604.04562
- How does the proposed dataset differ from the Reddit and Wikipedia datasets discussed in the last paragraph of the related work section? This should be explained.
- Page 3, paragraph "Conversational Flow": what is the maximum number of turns, if the minimum is 5?
- Page 3, paragraph "Knowledge Retrieval": how were the top 7 articles and first 10 sentences choices made? This seems arbitrary. Also, why wasn't the whole text used?
- Page 3, paragraph "Knowledge Selection and Response Generation": how do you deal with co-reference problems if you only ever select one sentence at a time? The same goes for the "Knowledge Attention" model described in Section 4.
- Page 3, paragraph "Knowledge Selection and Response Generation": how often do annotators choose "no sentence selected"? It would be interesting to see more such statistics about the dataset
- Section 4.2: did you run experiments for BPE encoding? Would be good to see as this is a bit of a non-standard choice.
- Section 4.2: it would be good to explain the Cer et al. 2018 method directly in the paper
- Section 4.2: is there a reference for knowledge dropout? Also, it would be good to show ablation results for this.
- Section 5.1: why did you choose to pre-train on the Reddit data? There should be some more in-depth description of the Reddit dataset to motivate this choice.
- Section 5.1: what is the setup you use for multi-task learning on SQuAD? Is it just a hard parameter sharing model, or?
- Section 5.3: as stated above, the human evaluation is a little bit underwhelming, both in terms of setup and results. I'd expect a more fine-grained way of assessing conversations by humans, and also an explanation of why the retrieval performer without knowledge was assessed as being on par with the retrieval transformer memnet.
- Section 5.3: I assume higher=better for the human scores? This should be made explicit.
- Section 5.3: Have others used the "F1 overlap score"? If so, cite.
- Section 5.3: I don't understand the argument that the human evaluation shows that humans prefer more natural responses. How does it show that?
- Section 5.3: The Wiki F1 score is kind of interesting because it shows to what degree the model uses knowledge. But the side-by-side comparison with the human scores shows that humans don't necessarily prefer chatbot models that use a lot of knowledge. I'd expect this to be discussed, and suggestions for future work to be made accordingly.
- Section 6: The paper ends a bit abruptly. It's be nice to suggest future areas of improvement.

---

> ### Author Response · Authors · 2018-11-27
> **Response**
>
> Thank you for your review and detailed feedback. We apologize if our answer is long, but you had a lot of questions! We have tried to answer them all and make necessary changes to the paper. Thank you for your constructive comments.
>
> - Missing reference for goal-oriented dialogue datasets: Wen et al. 2017, A Network-based End-to-End Trainable Task-oriented Dialogue System, https://arxiv.org/abs/1604.04562:
>     o  Thank you, we have added this citation.
>
> - How does the proposed dataset differ from the Reddit and Wikipedia datasets discussed in the last paragraph of the related work section? This should be explained:
>     o  Compared to the related datasets we describe in Section 2, our dataset provides specific and high-quality grounding as we ask the Wizard to author dialogue based on the given knowledge (so we know what to ground to later when training models). Further, we ask the Wizard to select which sentence the knowledge is from, giving even more fine-grained information. In those existing tasks the data used to ground was not accessible during dialogue collection, and thus may or may not be related. We believe this is why our task can lead to more successful models. Moreover, our task provides easier analysis, e.g. we can measure knowledge selection metrics. We have clarified this in the text.
>
> - What is the maximum number of turns:
>     o  There is no maximum number of turns, the two human speakers can continue to speak (but the crowdworker pay is fixed, so typically they only do this when they are really enjoying it). The maximum conversation length in the dataset is 23 utterances long. This has been clarified in the paper.
>
> - Page 3, paragraph "Knowledge Retrieval": how were the top 7 articles and first 10 sentences choices made? This seems arbitrary. Also, why wasn't the whole text used:
>     o  In order to keep crowdworkers from being overwhelmed, we chose for the worker to be exposed to no more than 15 different wikipedia articles at once (7 based on the Wizard's previous message, 7 based on the Apprentice's previous message, and 1 for the chosen topic), any higher was too much work to annotate. The first ten sentences translated to roughly the first and second paragraphs of the wikipedia topic article, which we felt would be ample information for the conversations. We did initially test varying amounts of sentences and articles, and we ultimately settled on these choices as they struck the best balance between keeping a conversation moving while also ensuring the Wizard had enough information to use in a response. Note this does not necessarily limit a model using more, but was simply the sentences shown to crowdworkers at training time.
>
> - Page 3, paragraph "Knowledge Selection and Response Generation": how do you deal with co-reference problems if you only ever select one sentence at a time? The same goes for the "Knowledge Attention" model described in Section 4:
>     o  We do not handle coreference in any special way, but it is part of the task. Annotators can read the sentences surrounding the one they click on, so they may make use of them. Further, the crowdworkers and the model were both provided titles of the Wikipedia articles as well as their content for each of the potential knowledge sentences. We observed that our generator model did learn to substitute this title in place of pronouns in certain cases. Making use of further context is not addressed in our particular models, but could be in future work.
>
> -  Page 3, paragraph "Knowledge Selection and Response Generation": how often do annotators choose "no sentence selected"? It would be interesting to see more such statistics about the dataset:
>     o  The Wizards choose “no sentence selected” around 6.2% of the time. We have provided additional details about the human annotation interface, topic selection, and information about the data in the Appendix.

---

> > ### Author Response · Authors · 2018-11-27
> > **Response Part 2**
> >
> >
> > -  Section 4.2: did you run experiments for BPE encoding? Would be good to see as this is a bit of a non-standard choice:
> >     o  We found modeling word piece tokens eased generation difficulty as it reduces the vocabulary size and quantity of rare words. It has been widely used in other sequence-to-sequence modeling tasks, such as machine translation and summarization. In datasets based on Wikipedia, there are often large quantities of rare words. Previous work has found BPE tokenization improves the model's ability to copy rare words, particularly entities (https://arxiv.org/abs/1711.05217). In early iterations of our models, we found that models without BPE tokenization could not produce UNK tokens. We contemplated implementing a copy-pointer mechanism (See et al., 2017), but found that BPE adequately addressed this problem for our purposes. We would be excited to see the effect of a copy mechanism in future work. Our new task and dataset clearly leave many avenues of research still open.
> >
> > - Section 4.2: it would be good to explain the Cer et al. 2018 method directly in the paper:
> >     o  Cer et al. 2018 propose using sum(vectors)/sqrt(sentence length), instead of of the mean, sum(vectors)/sentence length, in order to balance short and long sentences. We added this clarification to the paper.
> >
> > - Section 4.2: is there a reference for knowledge dropout? Also, it would be good to show ablation results for this:
> >     o  To our knowledge, knowledge dropout is unique to our paper, but it is similar to other techniques like token dropout (see e.g. https://arxiv.org/pdf/1709.03856.pdf). Table 4 contains a comparison with and without knowledge dropout; we emphasized this ablation in the text, and added a reference.
> >
> > - Section 5.1: why did you choose to pre-train on the Reddit data? There should be some more in-depth description of the Reddit dataset to motivate this choice:
> >     o  Mazare et al (https://arxiv.org/pdf/1809.01984.pdf) found success with pretraining on a large-scale dataset of conversations with personas extracted from Reddit dumps. They show that by pre-training on this Reddit data and finetuning on the Personachat dataset improved model accuracy by over 18%. We use the same procedure described in that paper.
> >
> > - Section 5.1: what is the setup you use for multi-task learning on SQuAD? Is it just a hard parameter sharing model, or:
> >     o  We multi-task with SQuAD by formulating the SQuAD task as a ranking one: namely, the model is tasked with finding the sentence(s) in the context paragraph that contain the correct answer. In this way we multi-task in the usual sense - by alternating training examples between the Wizard task and this re-formulated SQuAD task. This made minimal impact, but we hope future work will explore the relationship between knowledge grounded dialogue and QA tasks, which we have now added to the future work in the conclusion.

---

> > > ### Author Response · Authors · 2018-11-27
> > > **Response Part 3**
> > >
> > > - Section 5.3: as stated above, the human evaluation is a little bit underwhelming, both in terms of setup and results. I'd expect a more fine-grained way of assessing conversations by humans, and also an explanation of why the retrieval performer without knowledge was assessed as being on par with the retrieval transformer memnet:
> > >     o  Many papers with human evaluations only report one type of metric (quality/appropriateness), for example the following highly cited ones:   Vinyals & Le on the OpenSubtitles corpus (https://arxiv.org/pdf/1506.05869.pdf),  Li et al. (both https://arxiv.org/abs/1606.01541 and https://arxiv.org/pdf/1510.03055.pdf) and Liu et al. (https://arxiv.org/pdf/1603.08023.pdf) .  Other highly cited papers do not perform any human evaluations at all, e.g.   Lowe et al. on the Ubuntu corpus  https://arxiv.org/pdf/1506.08909.pdf  and Serban et al. on MovieTriples (http://www.aaai.org/ocs/index.php/AAAI/AAAI16/paper/download/11957/12160). Note, of the two recent works in knowledge grounding cited, Parthasarathi & Pineau (https://arxiv.org/abs/1809.05524) do not report human evaluation at all (only BLEU metrics), while Ghazvininejad et al. (https://arxiv.org/abs/1702.01932)  have two metrics: informativeness and appropriateness. Overall, a fine-gained way of assessing conversations is still an unsolved problem. What we do do in this work, in addition to automatic and human evaluations, is an error analysis in Appendix C.
> > >     o  For the retrieval models (memory vs. no memory),  note that our goal is to obtain both an engaging and knowledgable conversational agent. It is possible for a model to display a large amount of knowledge but to not be engaging at all, e.g. copy paste of sentences from Wikipedia. In our work we thus settled on two metrics for online human evaluation: engagingness and knowledge (Wiki F1) metrics, where we assume both should be maximized. In addition, we report a set of offline automatic evaluation metrics. We have now added text explaining this.
> > >
> > > - Section 5.3: I assume higher=better for the human scores? This should be made explicit:
> > >     o  Correct. We have updated the paper.
> > >
> > > -  “Have others used the F1 overlap score? If so, cite.''
> > >     o  Not that we know of.  We made this clearer in the paper.
> > >
> > > - Section 5.3: I don't understand the argument that the human evaluation shows that humans prefer more natural responses. How does it show that?
> > >     o  This is merely an observation that retrieval models tend to produce more “natural” responses— in the sense that they are retrieving from a set of human utterances— and for that reason humans can prefer these models to generative ones, which often produce short, generic, and repetitive responses (see https://arxiv.org/abs/1611.06216 and https://arxiv.org/pdf/1801.07243.pdf). Despite the fact that retrieval models sometimes produce erroneous or out of context responses, humans may rate these retrieval models more highly as they are simply easier to read. Nevertheless, we have removed this comment as it is hard to tease apart with complete certainty the clear reasons why one model is preferred over another here. We do however provide a detailed error analysis in Appendix C.
> > >
> > > - Section 5.3: The Wiki F1 score is kind of interesting because it shows to what degree the model uses knowledge. But the side-by-side comparison with the human scores shows that humans don't necessarily prefer chatbot models that use a lot of knowledge. I'd expect this to be discussed, and suggestions for future work to be made accordingly:
> > >     o  Our goal is to obtain both an engaging and knowledgable conversational agent. It is possible for a model to display a large amount of knowledge but to not be engaging at all, e.g. copy paste of sentences from Wikipedia. In our work we thus settled on two metrics for online human evaluation: engagingness and knowledge (Wiki F1) metrics, where we assume both should be maximized. In addition, we report a set of offline automatic evaluation metrics. Like in the entire domain of dialogue (not just knowledge-grounded models), evaluation is unfortunately still an active area of research.  We have added comments explaining this, as you suggest.
> > >
> > > - Section 6: The paper ends a bit abruptly. It's be nice to suggest future areas of improvement:
> > >     o  Thank you, we have enhanced our conclusion with future work.

---

### Official Review · AnonReviewer3 · 2018-11-02
**interesting task and dataset**

**Rating:** 6
**Confidence:** 5

**Review:**

This paper collects a new annotated dataset for knowledge grounded dialog task. The proposed models combine two recent neural networks, Memory Net and Transformer, for the purpose of the task. I highly appreciate the efforts to collect such a precious dialog dataset for the community. Also, the setup in data collection actually narrows down the scope of chitchat dialog into a specific topic by grounding it to a set of knowledge.

Here are summaries of my concerns and questions about the paper.

# applicability of the knowledgeable bot
What is the basic motivation of this work? Once you develop a chatbot that can produce a response grounded by knowledge, how could it be applied to real-world applications? Are you trying to teach a student who is looking for more knowledge about a topic? If so, you should be more careful about what knowledge the student (or apprentice in the paper) knows or don’t know about the topic and how their knowledge models dynamically change over the chat. Otherwise, the proposed model seems a simple knowledge retrieval model given the dialog context. Would you please provide motivations of the work?

# No explicit goal of a dialog makes the chat divergent and open-ended
Without a specific goal given to the annotators or a restriction in the instruction, a dialog in the current setting might diverge beyond the context. For example, if an apprentice says about her/his personal opinion about the topic (e.g., I hate the Gouda cheese) or past experience (e.g., I went to a music festival by Michael Jackson 23 years ago), then how do you control the chat between two annotators or how do you train a model not to pay much attention on out-of-topic utterances?

# Lack of further analysis of the dataset
Data collection part itself seems to be the biggest contribution to this work. Why don’t you bring one of real dialog example in Figure 3 to the main paper and say more about it? For example, what other interesting applications can you develop on this dataset?

Compared to the Wizard, the role of apprentice seems unclear to me. I found from the examples in Figure 3 that most of the apprentices’ responses are a follow-up question about the knowledge, a personal agreement or feeling or their preference. Do you have any post analysis on the types of responses from the apprentices so highlighting utilities of the dataset in a real application?

# Some questions on data collection
Do you have any incentive mechanism to make annotators more engage in the dialog?
Did you filter out some bad dialogs? Then, how did you measure the quality of a dialog?
How do you penalize bad annotators that often make aggressive words or don’t follow the instruction you set up?

# A question on the model
Compared to previous works such as (Zhang at al., ACL18), the proposed model seems to have the only replacement with Transformer encoder and a loss term for knowledge selection. Have you tried another way of dealing with the knowledge part? For example, a ranking loss might be better than the attention.

# Questions on the Experiment section
Any experiment to show the effect of different \lambda value in the loss of the generative model?

When you evaluate the generative model, have you also tried other automatic metrics such as BLEU instead of only PPL and Unigram-F1? For this task, the possible response grounded by the topic+knowledge might be too diverse to measure though. Could you possibly add some constraints to the annotators to do some clear tasks over the dialog so you can systematically evaluate the dialog w.r.t the constraint? Otherwise, evaluation of this task seems to be mostly the same as chitchat systems.

In Table 5, human evaluators only measure the likeness of the dialog which seems very naive. Why don’t you measure whether the apprentice gets new knowledge of which s/he didn’t know before, whether the knowledge provided from the model was informative, whether the dialog was fun and engaging or more? The current human evaluation seems very weak though.

This might be an auxiliary question: have you tried to train the model for apprentice and make two models chat with each other? How does the chat look like then?

---

> ### Author Response · Authors · 2018-11-27
> **Response**
>
> Thank you for your review and detailed feedback, we appreciate the constructive comments. We apologize if our answer is long, but you had a lot of questions! We have tried to answer them all and make necessary changes to the paper.
>
> - Real application:
>     o  This task is not meant to be solely a diagnostic dataset, but a basis for a knowledgeable conversational agent that can talk about any knowledge that is in Wikipedia. We are interested in this task, because an agent that is both knowledgeable and can converse with humans in an engaging way is one of the goals of AI.  A successful system could engage with real people (not just paid crowdworkers). In our work, the goal is to chat freely about the topic, i.e. a chitchat task.  No, we do not aim to make an educational tool as you mention, although others could use our work as a pre-training for such a task perhaps. However, we respectfully disagree that our models are “a simple knowledge retrieval model given the dialog context”. Please see e.g. Figures 2 and 4 to show that in the best cases, where our modeling works very well,  particularly the E-book, toga party and Arnold Schwarzenegger examples the agent can be very conversationally engaging — it both uses knowledge, but also clearly replies and follows the conversation of  the human partner, producing engaging conversations as measured in human evaluations.
>
> - Test-bed for state-of-the-art dialogue models:
>     o  Separately, our task is also a challenging setup to develop models that can actually talk in a knowledgeable way to humans. They must have a memory (and be able to retrieve knowledge from it), to be able to select that knowledge and converse convincingly with respect to the dialogue context. This combines a lot of the current research threads into a single challenging task where grounded knowledge can clearly be leveraged, due to the way the data was collected.
>
> - “No explicit goal of a dialog makes the chat divergent and open-ended”:
>     o  Yes! This is one of the challenges of real dialogue, and our dataset as well. Because our data set collection involves an in-the-loop knowledge retrieval at every dialogue turn during both data collection (and for models working on our task) the human Wizard is able to ground their conversation with knowledge from Wikipedia. That is, if they started talking about cheese, those topics will appear from the retrieval over Wikipedia, but if they switch to Michael Jackson, that will appear too. They are not locked into the original topic, just as in a natural conversation. This is what can make our models more feasibly useful for real chat in an application.
>
> - Additional details about the Apprentice:
>     o  The Apprentice is a completely unconstrained human, playing the role of a curious learner, eager to chat. Their stated goal is to go into depth about a chosen topic that interests themselves or their partner, while keeping the conversation engaging and fun. We observed Apprentices saying statements, asking questions and answering questions, as shown in the examples in Appendix A.2. Assuming that a question contains a question mark or begins with 'how', 'why', 'who', 'where', 'what' or 'when', in the dataset Apprentices ask questions in 13.9% of training set utterances, and answer questions (i.e., the Wizard has asked a question) 39.5% of the time, while saying new statements (neither asking nor answering a question) 49.3% of the time. (Note those percentages don't add up to 100 because a question may be answered with another question.) That is, overall, the Apprentice maintains a balanced set of dialogue acts. We made this clearer in the main text, and added details to the appendix.
>
> - “Do you have any post analysis on the types of responses from the apprentices so highlighting utilities of the dataset in a real application?”:
>     o  Please see the answer above which is now added to the paper.  We also note that we do provide an analysis of our models in Appendix C, which involves models talking to human Apprentices.
>     o  The dataset statistics are Table 1, examples from human-human conversations are in Fig 3 and examples dialogues of different models are in 2,4 & 5. There is unfortunately little room left in the main paper for more, hence they are in the appendix. We felt it was important to highlight the successes of the models, as to our knowledge there is scant evidence of models working this well in open-domain chitchat before.  Hence, the majority of our analysis has been on the modeling side (see Appendix C).

---

> > ### Author Response · Authors · 2018-11-27
> > **Response Part 2**
> >
> > - “Data collection part itself seems to be the biggest contribution to this work.”:
> >     o  To our knowledge there is little or no evidence of models working this well in open-domain chitchat before, and we believe that is the main contribution of this work. Our results stem from contributing models that can effectively leverage knowledge (Transformer Memory Networks) that are trained on a supervised grounded supervised dataset — i.e. the new Wizard of Wikipedia task. So you are right that the new grounded dataset is important, but note also that the baseline models from the literature we tried also failed to do this.
> >
> > - “For example, what other interesting applications can you develop on this dataset?”:
> >     o We believe the application as described is the application of this dataset. However, if in future work other researchers find other uses for it, then that is great as well. But we are focused on the task of open-domain knowledgeable conversations.
> >
> > - “Some questions on data collection”:
> >     o  We provided bonuses to high quality annotators manually. We discarded poor quality conversations through a mixture of manual inspection and automatic tests where external knowledge was not used in at least 2 Wizard turns (which was determined via at least 3 words of non-stop word overlap between checked sentence and Wizard message), and subsequently did not allow those workers to have further conversations. There were also a few annotators whom others would individually report for various reasons. Finally, we also implemented an offensive language detection system to auto-reject/block workers who used such language.
> >
> > - A question on the model compared to previous works such as (Zhang at al., ACL18), the proposed model seems to have the only replacement with Transformer encoder and a loss term for knowledge selection. Have you tried another way of dealing with the knowledge part? For example, a ranking loss might be better than the attention.
> >     o  Yes, we believe the Transformer encoder is a better choice than a bag of words used in Zhang et al., 2018; we do compare explicitly to a bag of words encoder in Tables 2 and 3. We did not experiment with other alternative attention or ranking mechanisms for the knowledge choices, but there are many avenues to explore in future work.
> >
> > - Questions on the Experiment section. Any experiment to show the effect of different \lambda value in the loss of the generative model?
> >     o  We treat lambda as a hyperparameter and tune using the validation set using grid search. An informal trial-and-error search showed valued closer to 1 worked better. During the final tuning, we tried values {0.8, 0.9, 0.95}.
> >
> > - When you evaluate the generative model, have you also tried other automatic metrics such as BLEU instead of only PPL and Unigram-F1? For this task, the possible response grounded by the topic+knowledge might be too diverse to measure though.
> >     o  We experimented with BLEU-4 as an automatic metric, but concluded it was a poor choice which did not reward diverse responses, particularly since responses usually do not have strong n-gram alignments like in machine translation. We found BLEU was unable to distinguish strong models from weak models in a way that corresponded to our own interactions with the models, and as such, omitted these results from the paper.
> >
> > - “Could you possibly add some constraints to the annotators to do some clear tasks over the dialog so you can systematically evaluate the dialog w.r.t the constraint?”:
> >     o  indeed our task _is_ a chitchat task, in line with other chitchat tasks.  Your goal-oriented proposal (which is somewhat open-ended, so would have to be nailed down) is beyond the scope of this work, however we think is an interesting & exciting extension we hope others could try to pursue in the future in order to link chitchat and goal-oriented dialogue together.

---

> > > ### Author Response · Authors · 2018-11-27
> > > **Response Part 3**
> > >
> > > - In Table 5, human evaluators only measure the likeness of the dialog which seems very naive. Why don’t you measure whether the apprentice gets new knowledge of which s/he didn’t know before, whether the knowledge provided from the model was informative, whether the dialog was fun and engaging or more? The current human evaluation seems very weak though.
> > >     o  Human evaluation is notoriously hard, and in fact many questions asked to humans either have very little inter-annotator agreement, e.g. when asking about specificity or background knowledge, (see https://arxiv.org/pdf/1708.07149.pdf, appendix A (https://arxiv.org/pdf/1708.07149.pdf)), or else are so correlated with each other they do not give new information and are hence not really connected to the intent of the original question, e.g. appropriateness v.s. topicality (see same cite). That paper recommends only asking humans for one score.
> > >     o  Indeed, many papers with human evaluations only report one type of metric (usually, quality/appropriateness), for example the following highly cited ones:  Vinyals & Le on the OpenSubtitles corpus (https://arxiv.org/pdf/1506.05869.pdf),  Li et al. (both https://arxiv.org/abs/1606.01541 and https://arxiv.org/pdf/1510.03055.pdf) and Liu et al. (https://arxiv.org/pdf/1603.08023.pdf) .  Yet other highly cited papers do not perform any human evaluations at all, e.g.  Lowe et al. on the Ubuntu corpus  https://arxiv.org/pdf/1506.08909.pdf  and Serban et al. on MovieTriples (http://www.aaai.org/ocs/index.php/AAAI/AAAI16/paper/download/11957/12160).
> > >     o  Note of the two recent works in knowledge grounding cited, Parthasarathi & Pineau (https://arxiv.org/abs/1809.05524) do not report human evaluation at all (only BLEU metrics), while Ghazvininejad et al. (https://arxiv.org/abs/1702.01932)  have two metrics: informativeness and appropriateness. Overall, a fine-gained way of assessing conversations is still an unsolved problem. What we do do in this work, in addition to automatic and human evaluations, is an error analysis in Appendix C.
> > >
> > > - Training/trying two apprentice models:
> > >     o  We have not tried to train an Apprentice model,  but that is interesting future work that could be pursued. Those models would likely contain much less knowledge.

---

### Official Review · AnonReviewer2 · 2018-11-02
**Good work**

**Rating:** 7
**Confidence:** 4

**Review:**

This work proposes a brand new dataset to fill in the vacancy of current conversational AI community, specifically the introduced dataset aims at providing a platform to perform large-scaled knowledge-grounded chit-chat. Overall, the dataset is well-motivated and well-designed, its existence will potentially benefit the community and inspire more effective methods to leverage external knowledge into dialog system. Besides, the paper also utilizes many trending models like Transformers, Memory Networks, etc to ensure the state-of-the-art performance. The clear structure and paragraphs also makes the paper easy to read and follow.

Here are some questions I want to raise about the paper:

1. First of all, the design of the conversation flow though looks reasonable, but it is pretty uncommon for a human to ground his/her every sentence on external knowledges. Therefore, it would probably be better to introduce some random ungrounded turns into the conversation to make it more humanlike.

2. Secondly, the whole framework is based on many modules and every one of them are prone to error. I’m afraid that such cascaded errors will accumulate and lead to compromised performance in the end. Have you thought about using REINFORCE
algorithm to alleviate this issue?

3. Finally, it would be better to introduce some noisy or adversarial apprentice to raise unrelated turns and see how the system react. Have you thought about how to deal with such cases?

---

> ### Author Response · Authors · 2018-11-27
> **Response**
>
> Thank you for your review and detailed feedback. We address your 3 points below:
>
> - “it is pretty uncommon for a human to ground his/her every sentence on external knowledges”:
>     o  In fact in our task, at any time in the dialogue the Wizard can choose “no sentence used instead”, as stated in the paper, but we have made this clearer (adding it in two places). We do agree that the Wizard task is not completely natural for a human as we are trying to make the human conversationalist help to train our bot maximally by grounding their sentences so that we can learn how to ground — for this reason we ask the human to read Wikipedia sentences and use them if possible rather than their own personal knowledge (which the model cannot retrieve). It is this setup that we believe makes our dataset so useful for knowledge grounded dialogue model training compared to existing datasets.
>
> - REINFORCE for learning the whole system:
>     o  Indeed our methods do use at least two modules: one for knowledge retrieval (searches over all of Wikipedia), and one that combines knowledge selection and generation/retrieval. Training the parts of the system together, e.g. by REINFORCE is a great idea for future work. We have added this to the future work section of the conclusion.
>
> - Noisy or adversarial apprentice:
>     o  Again, making the systems more robust is also a good direction for future work. Our task will be made publicly available for researchers to try such improved follow-up techniques.

---

### Meta-Review · Area_Chair1 · 2018-11-05
**Interesting dataset and evaluation framework**

**Confidence:** 4
**Recommendation:** Accept (Poster)

**Metareview:**

The paper proposes a new dataset for studying knowledge grounded conversations, that would be very useful in advancing this field. In addition to the details of the dataset and its collection, the paper also includes a framework for advancing the research in this area, that includes evaluation methods and baselines with a relatively new approach.
The proposed approach for dialogue generation however is a simple extension of previous work by (Zhang et al) to user transformers, hence is not very interesting. The proposed approach is also not compared to many previous studies in the experimental results.
One of the reviewers highlighted the weakness of the human evaluation performed in the paper. Moving on, it would be useful if further approaches are considered and included in the task evaluation.

A poster presentation of the work would enable participants to ask detailed questions about the proposed dataset and evaluation, and hence may be more appropriate.